# Therapy of Patients with Cardiac Malposition

**DOI:** 10.3390/children10040739

**Published:** 2023-04-17

**Authors:** P. Syamasundar Rao

**Affiliations:** Children’s Heart Institute, Children’s Memorial Hermann Hospital, McGovern Medical School, University of Texas-Houston, 6410 Fannin Street, UTPB Suite # 425, Houston, TX 77030, USA; p.syamasundar.rao@uth.tmc.edu; Tel.: +1-713-500-5738; Fax: +1-713-500-5751

**Keywords:** dextrocardia, levocardia, dextroposition, Blalock–Taussig shunt, pulmonary artery banding, bidirectional Glenn operation, Fontan operation, coronary artery disease, arrhythmia

## Abstract

Positional abnormalities per se do not require treatment, but in their place, the accompanying pulmonary pathology in dextroposition patients and pathophysiologic hemodynamic abnormalities resulting from multiple defects in patients with cardiac malposition should be the focus of treatment. At the time of the first presentation, treating the pathophysiologic aberrations caused by the defect complex, whether it is by improving the pulmonary blood flow or restricting it, is the first step. Some patients with simpler or single defects are amenable to surgical or transcatheter therapy and should be treated accordingly. Other associated defects should also be treated appropriately. Biventricular or univentricular repair dependent on the patient’s cardiac structure should be planned. Complications in-between Fontan stages and after conclusion of Fontan surgery may occur and should be promptly diagnosed and addressed accordingly. Several other cardiac abnormalities unrelated to the initially identified heart defects may manifest in adulthood, and they should also be treated.

## 1. Introduction

In a previous paper on the “Diagnosis of Dextrocardia with a Pictorial Rendition of Terminology and Diagnosis”, a detailed discussion of the diagnosis of dextrocardia including other cardiac malpositions was presented [1]. In that review, a pictorial presentation of terminology and diagnosis was made. Following exclusion of dextroposition, segmental analysis is undertaken examining the visceroatrial situs, ventricular localization, atrioventricular connections, great artery relationship, and conotruncal alignment. Electrocardiogram, chest X-ray, echocardiogram, and other imaging studies, including selective cineangiography, are utilized for this assessment. Once the atrial, ventricular, and great artery sites are identified, septal defects and obstructive lesions of cardiac valves and blood vessels are diagnosed using the conventional cardiovascular evaluation methodology. Treatment of cardiac malposition was not included in that paper [1] because of space limitations. In the current paper, therapy of cardiac malposition is reviewed.

## 2. Therapy

Abnormal positions themselves do not require therapy; however, the associated boney and pulmonary abnormalities in dextroposition subjects and hemodynamic anomalies caused by multiple associated defects in patients with cardiac malposition should be treated. The management of dextroposition and dextrocardia, including other cardiac malpositions, will be reviewed separately. Discussion of management of other cardiac malpositions, namely, ectopia cordis and pericardial defects, is not included in this paper.

## 3. Management of Dextroposition

In patients with dextroposition secondary to a thoracic cage boney abnormality, no treatment is necessary because the heart is usually just displaced without any compression of the cardiac chambers. On extremely rare occasions, when obstruction of a given cardiac chamber occurs [2], treatment may be warranted to relieve the obstructive element. In patients with cardiac displacement due to empyema, pneumothorax, congenital pulmonary cyst, lobar emphysema, or left-sided diaphragmatic hernia displacing the heart to the right side and collapse of the right lung, hypoplastic right lung, or right pneumonectomy pulling the heart to the right, the primary lung abnormality should be addressed as detailed in standard textbooks of pediatrics, pediatric surgery, and respiratory system disorders and is not reviewed here. In subjects with scimitar syndrome, the accompanying partial anomalous pulmonary venous return, defects in the atrial septum, and an abnormal systemic vascular connection to the sequestered pulmonary segment should be addressed surgically or by transcatheter methodology, as deemed appropriate, [3,4,5,6] and no treatment of cardiac displacement is necessary.

## 4. Management of Cardiac Malposition

Positional abnormalities of the heart themselves do not require any therapy, but, in their place, the accompanying heart defects and the ensuing hemodynamic abnormalities should be treated and are reviewed hereunder. The clinical scenarios and their management options are listed in a tabular form in Table 1.

### 4.1. Management at Initial Presentation

Patients with complex heart defects present with symptomatology early in the newborn period while subjects with simpler and milder heart defects may come to light later in infancy, childhood, and adulthood. Treatment at first presentation of neonates is similar to that utilized for other cyanotic congenital heart defects (CHDs) and is described elsewhere [7,8] and includes avoiding hypothermia, preserving neutral thermal setting, detection and appropriate treatment of hypocalcemia and hypoglycemia, monitoring the acid–base balance, treating metabolic acidosis with NaHCO_3_, and managing acidosis secondary to respiratory problems with suction, tracheal cannulation, and assisted ventilation as suitable. In patients with suspected ductal dependent lesions, intravenous prostaglandin E_1_ (PGE_1_) infusion [9,10,11,12] should begin while waiting for the final diagnosis.

Following stabilization of the baby, treatment should focus on the pathophysiologic anomaly (pulmonary oligemia and pulmonary plethora) produced by the associated cardiac defects that many of the cardiac malposition infants have. Pulmonary oligemia should be addressed at presentation to prevent both short-term adverse effects such as hypoxemia, progressive acidosis, and death and long-term consequences, namely, the development of clubbing of fingers and toes, cerebrovascular accidents, relative anemia, severe polycythemia, paradoxical embolism, brain abscess, coagulation abnormalities, hyperuricemia, uric acid nephropathy, and gout. Pulmonary plethora should be tackled to avoid short-term effects, namely, congestive heart failure (CHF) and death as well as long-term consequences, namely, the development of pulmonary hypertension, progressive dilation of the left ventricle, and pulmonary vascular obstructive disease (PVOD). In addition, addressing both pulmonary oligemia and pulmonary plethora in a timely manner is likely to help normal pulmonary alveolar development.

In infants with decreased pulmonary blood flow secondary to severe narrowing or atresia of the pulmonary artery or valve and a big VSD or a single ventricle, administration of PGE_1_ starts at an initial dosage of 0.05 to 0.1 μg/kg/min, which may be gradually decreased to 0.015 to 0.02 μg/kg/min. Following stabilization of the infant, a more permanent provision of blood flow to the pulmonary circuit should ensue, usually by a modified Blalock–Taussig (BT) anastomosis [13,14,15] (Figure 1 and Figure 2). Alternative solutions are ductal stenting [16,17,18] and balloon pulmonary valvuloplasty [19], where applicable.

Babies with elevated pulmonary blood flow and CHF should receive decongestive treatment and then surgery to repair the defect if amenable to surgical correction as an infant. If not, pulmonary artery banding [22,23,24] is undertaken (Figure 3 and Figure 4) to decrease the pressures and blood flow in the pulmonary circuit and avoid the development of PVOD. In this scenario, total surgical correction is planned to be performed later. Nevertheless, since there is an elevated pulmonary vascular resistance (PVR) when the baby is born and a slow regression of the PVR in patients with large interventricular communication, most of these neonates may not go into CHF early in life. These infants should be diligently followed and treated to prevent the onset of PVOD.

Neonates with the total anomalous pulmonary venous return (TAPVR) with an obstruction to the pulmonary venous return should have immediate surgical correction [26,27,28,29]. It should be noted that some newborn infants with markedly reduced pulmonary blood flow may manifest TAPVR only after performing BT shunting. Currently, improved echo techniques facilitate accurate identification of TAPVR at the time of the initial presentation.

Other problems, namely, interatrial obstruction, interventricular obstruction, aortic arch interruption, and aortic coarctation may be seen at presentation or may develop later [25] and should be addressed with transcatheter or surgical therapy as appropriate.

Subjects with simpler and single defects such as pulmonary valvar stenosis [30], atrial [31,32,33,34,35] or ventricular [36,37] septal defects and patent ductus arteriosus, though rare, should be addressed with either catheter-based [31,32,33,34,35,36] or surgical [37] procedures, as deemed appropriate.

### 4.2. Additional Surgery

After initial palliation, as reviewed above, a methodical review of all the available imaging studies should be performed to assess the feasibility of undertaking biventricular repair [38,39,40,41]. If biventricular repair is not possible, consideration for single ventricle palliation via the Fontan route [42,43,44,45] should be given. Currently, the Fontan procedure is accomplished using the concept of total cavopulmonary connection (TCPC) proposed by de Leval et al. [46] in a staged fashion [47]. An extracardiac conduit [48,49] with fenestration [50,51,52] is used for completion of the Fontan operation Subsequently, device occlusion of the fenestration [51,53,54,55,56] is undertaken.

#### 4.2.1. Stage I

This stage is as described in the “Management at Initial Presentation” segment.

#### 4.2.2. Stage II

This comprises bidirectional Glenn operation [57,58,59,60,61,62], diverting the superior vena caval blood flow into the pulmonary circuit (Figure 5 and Figure 6) at about six months of age.

#### 4.2.3. Stage IIIA

This comprises re-routing the inferior vena caval blood flow to the pulmonary circuit with a non-valved conduit [48,49] (Figure 7 and Figure 8) along with fenestration [50,51,52] (Figure 8 and Figure 9) a year or two after Stage II.

#### 4.2.4. Stage IIIB

Stage IIIB comprises device closure of the fenestration (Figure 10 and Figure 11); this is performed six to twelve months following Stage IIIA [51,53,54,55,56].

## 5. Management of Noncardiac Issues

As mentioned above, some of the babies with cardiac malposition, particularly those with isolated dextrocardia and isolated levocardia, are likely to have heterotaxy, and these patients with asplenia/polysplenia syndrome usually have a deficient splenic function. Consequently, such infants are at a greater risk of bacterial infections [64,65]. Throughout infancy, these babies should be treated with prophylactic doses of ampicillin/amoxicillin. During childhood, the preventive regimen may be switched to oral penicillin. Furthermore, there is a high risk of occurrence of gastrointestinal obstruction and, therefore, prophylactic Ladd’s procedure [66,67] is recommended at most institutions caring for such babies to prevent the development of an obstruction of the gastrointestinal tract.

## 6. Interstage Issues

In patients with single ventricles, both the pulmonary and systemic circulations work in parallel in place of the normal in-series circulation. Consequently, a finite equilibrium connecting both circulations should be preserved to ensure an adequate flow into both systemic and pulmonary circuits [25,68]. In case such an equilibrium cannot be preserved, significant morbidity and mortality may ensue in these susceptible babies. Mortality in-between stages has been reported, which varies from 5 to 15% [25,69,70]. The identified causes of interstage mortality are obstructed patent foramen ovale (PFO), aortic coarctation, aortic arch interruption, narrowing of the pulmonary arteries, atrioventricular valve insufficiency, obstruction of the surgically created aortopulmonary shunts, and intercurrent illnesses [69,70]. The stated interstage mortality occurs more often in-between Stages I and II than in-between Stages II and III. Although these studies [69,70] mostly involve hypoplastic left heart syndrome subjects, these findings are equally relevant to patients with a single ventricle seen in malposition of the heart. Strategies to avert and manage interstage issues are timely assessment by careful history and physical examination, including echocardiography and other imaging investigations, namely, magnetic resonance imaging (MRI) and computed tomography (CT), to detect the anomalies listed in the preceding section and provide a suitable management of these problems to avert/decrease the mortality and morbidity [20,25,71,72]. Intercurrent illnesses resulting in dehydration, acidosis, or high temperature may interrupt the balance in-between the systemic and pulmonary circuits, and the infants could develop serious illness [71,72]. Rapid attention to treat intercurrent illnesses is important [25,71,72]. Further review of the methods of treatment of interstage issues is beyond the scope of this paper and the involved persons may refer to other reviews [20,25,73].

## 7. Postintervention Complications

Several complications were reported in subjects who had a Fontan operation. The reported complications were arrhythmias, obstructed Fontan pathways, residual intra-/extracardiac shunts, cyanosis, thromboembolism, transient ischemic attacks, cerebrovascular accidents, collateral vessel formation, congestion of the systemic venous circuit, and protein-losing enteropathy [63,74,75,76]. Coincidentally, such problems happen less often in babies who had undergone the presently utilized staged TCPC with an extracardiac conduit than in subjects who had previously utilized the atriopulmonary anastomosis type of Fontan surgery. Clinical and echocardiographic evaluation for the existence of the abovelisted complications during follow-up and prompt treatment by either surgical or transcatheter methodology, as deemed appropriate, should be pursued. A review of these management options is beyond the scope of this paper; this information may be found elsewhere [63,73,75,76].

## 8. Management of Cardiac Malposition Patients during Adulthood

Adult patients with cardiac malposition, apart from the postintervention problems reviewed in the aforementioned sections, may also develop problems akin to those seen in normal (levocardia) adults. Management of these issues is reviewed in this section.

### 8.1. Coronary Artery Disease

Coronary artery obstruction and myocardial infarction do occur in cardiac malposition patients; the prevalence of coronary problems is generally thought to be comparable to that seen in the general population [77]. It is necessary to record right chest leads to detect the ST segment elevation to identify myocardial infarction [78,79,80,81]. Percutaneous coronary intervention (PCI) [82,83,84,85,86,87,88,89] or surgical coronary artery bypass (CAB) [90], as deemed appropriate, should be performed promptly. Transradial approach for PCI has also been used successfully [91,92]. Immediate results of both PCIs and CAB [77,82,83,84,85,86,87,88,89,90,91,92] appear to be beneficial while long-term results have not been documented.

### 8.2. Supraventricular Arrhythmias

Supraventricular arrhythmias also occur in dextrocardia patients, and it is generally thought that the incidence of these arrhythmias is not substantially different from that observed in the normal population (levocardia). Atrial fibrillation/flutter appear to be the most common among these arrhythmias while supraventricular tachycardia (SVT), including atrioventricular reentrant tachycardia (AVRT) and SVT associated with WPW (Wolff–Parkinson–White) syndrome, can also occur. Arrhythmias are usually treated with antiarrhythmic medications and patients without adequate control are subjected to catheter ablation procedures (cryoablation or radiofrequency (RF) ablation). The feasibility of catheter ablation procedures for atrial fibrillation/flutter [93,94,95,96,97,98,99,100], supraventricular tachycardia [101,102,103], including AVRT [101] and WPW syndrome [104], was demonstrated. Patients with an interrupted inferior vena cava should undergo the procedure using the jugular venous approach. Integration of echocardiograms, cardiac CT, three-dimensional (3D) imaging, and 3D reconstruction are necessary to accomplish catheter ablation procedures.

Subjects with refractory arrhythmias may require the Maze procedure which can also be successfully conducted in dextrocardia patients [105,106].

### 8.3. Complete Atrioventricular Block or Sick Sinus Syndrome

Cardiac malposition patients with complete heart block [107] or sick sinus syndrome [108,109] require pacemaker therapy. The feasibility of implanting a dual-chamber pacemaker [107,108,109] in these patients is well-demonstrated. Leadless pacemaker implantation [110,111,112] can also be performed.

### 8.4. Prevention

#### 8.4.1. Prevention of Sudden Death

Some patients with dextrocardia have a higher propensity for sudden cardiac death; such patients may need implantable cardioverter–defibrillators (ICD). While transvenous ICDs are the first-line therapy, transvenous access may be difficult because of congenital venous anomalies or acquired venous obstruction. In such patients, subcutaneous ICDs (S-ICDs) (Emblem A219; Boston Scientific, Marlborough, MA, USA) may be implanted. The reported experience with this form of therapy is encouraging [113,114,115,116,117].

#### 8.4.2. Prevention of Embolic Episodes

Subjects who have atrial fibrillation/flutter are at risk of developing embolic episodes and, therefore, initial anticoagulation and subsequent occlusion of the left atrial appendage are recommended. While closure of the appendage of the left atrium is challenging in dextrocardia subjects, it can be successfully accomplished [118,119,120].

### 8.5. Aortic Valve Disease

Aortic valve (AV) abnormalities causing severe aortic regurgitation or severe stenosis may occur in dextrocardia patients [121,122,123,124,125,126]. Simpler balloon aortic valvuloplasty or surgical repair may be offered in younger populations. Transcatheter [121,122,123,124,125] or surgical [126] replacement of the AV may be offered for the elderly to address AV abnormalities [121,122,123,124,125,126].

### 8.6. Mitral Valve Disease

Mitral valve abnormalities producing severe mitral regurgitation have been seen in dextrocardia. Surgical repair [127] or replacement [128,129,130] of the mitral valve can be safely performed [127,128,129,130].

Rheumatic fever appears to occur in dextrocardia patients as well, resulting in mitral stenosis. When mitral stenosis is severe, surgical commissurotomy [131] or balloon mitral valvuloplasty with either conventional balloons or Inoue balloons [132,133,134,135,136,137,138,139,140] could be successfully performed.

### 8.7. Hypertrophic Cardiomyopathy and Subaortic Obstruction

Hypertrophic cardiomyopathy (HCM), while rarely reported in patients with cardiac malposition, has been described [141,142,143]. Some patients with HCM may have a severe obstruction of the outflow tract of the left ventricle [144]. If betablocker therapy does not relieve the left ventricular outflow tract (LVOT) obstruction, surgical myectomy or alcohol septal ablation (ASA) [144] may become necessary. In this report [144], the authors were able to reduce the LVOT gradient from 136 to 50 mm Hg along with symptomatic improvement. The presented case illustrates the feasibility of performing ASA in dextrocardia patients.

## 9. Summary and Conclusions

Addressing pulmonary pathology in patients with dextroposition as per the guidelines established in standard textbooks is appropriate. In cardiac malposition, treatment of the pathophysiologic abnormality produced by multiple defects at the time of initial presentation by either increasing the pulmonary blood flow or restricting it is first undertaken. Other cardiac defects are also addressed to optimize the care. The feasibility of biventricular repair should be explored, and if that is not feasible, a single ventricular repair option (Fontan) should be offered. Antibiotic prophylaxis and prophylactic Ladd’s procedure are prescribed in heterotaxy patients. Complications following initial palliative procedures and subsequent Fontan surgery may occur, and these should be promptly detected and treated in a timely manner. Adult patients with dextrocardia, in addition to postintervention complications, may develop other disease entities, similar to those seen in the normal population, and these should be treated with either surgical or transcatheter methodology, as deemed appropriate. It is concluded that most of the problems encountered in patients with dextroposition and dextrocardia can be effectively treated with the currently available therapeutic modalities.

## Figures and Tables

**Figure 1 children-10-00739-f001:**
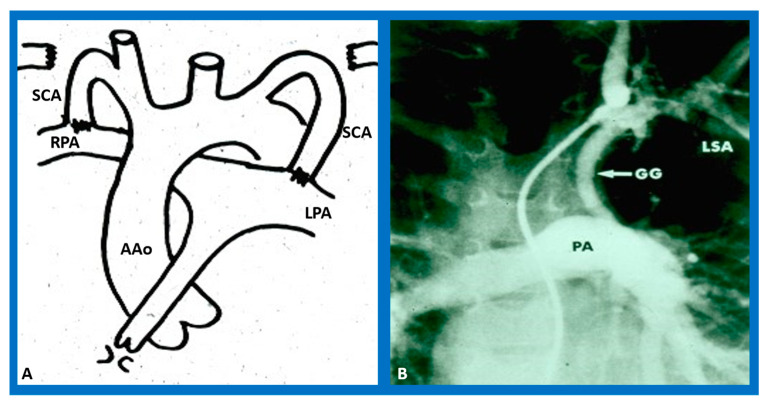
(**A**) The diagram drawn by hand by Dr. Taussig herself demonstrating the idea of the Blalock–Taussig shunt [9] indicating the connection of subclavian arteries (SCAs) to the right (RPA) or left (LPA) pulmonary artery, correspondingly. (**B**) Selected cineangiographic frame of a modified Blalock–Taussig anastomosis with a Gore-Tex graft (GG) connecting the left subclavian artery (LSA) with the pulmonary artery (PA) [10]. This figure reveals a wide-open Blalock–Taussig anastomosis and good opacification of the PA. AAo, ascending aorta. Reproduced from [20].

**Figure 2 children-10-00739-f002:**
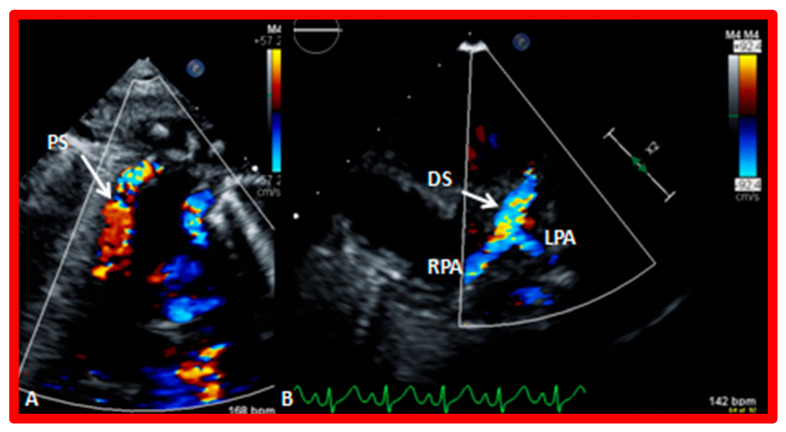
Echo–Doppler recordings acquired with a transducer positioned in the suprasternal notch view illustrating the proximal portion of the shunt (PS) by color Doppler (**A**). With a somewhat separate angulation (**B**), the distal part of the shunt (DS) is shown with flow into the right (RPA) and left (LPA) pulmonary arteries. Reproduced from [21].

**Figure 3 children-10-00739-f003:**
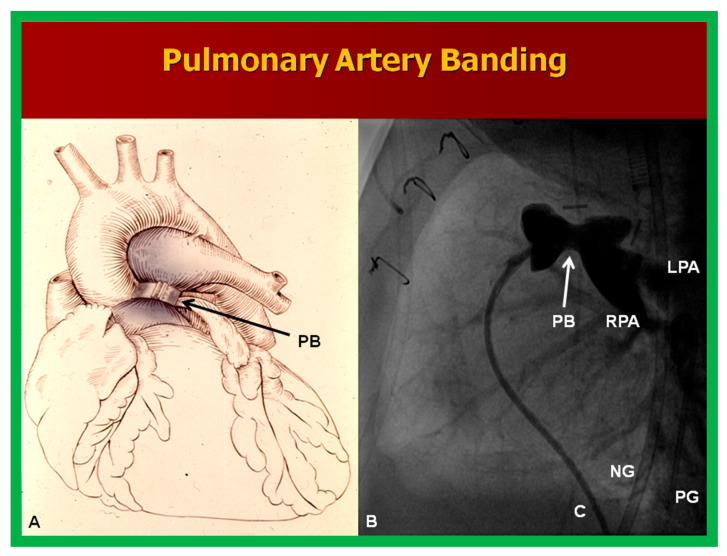
(**A**) A diagram illustrating pulmonary artery banding (PB) for patients with a severely increased blood flow into the pulmonary circuit and severe heart failure. (**B**) Cineangiographic image of the pulmonary artery angiogram in the lateral projection illustrating the narrowing of the pulmonary artery shown with an arrow in a baby who had surgical PB. Note: catheter (C); left pulmonary artery (LPA); nasogastric tube (NG); pigtail catheter (PG); and right pulmonary artery (RPA). Reproduced from [25].

**Figure 4 children-10-00739-f004:**
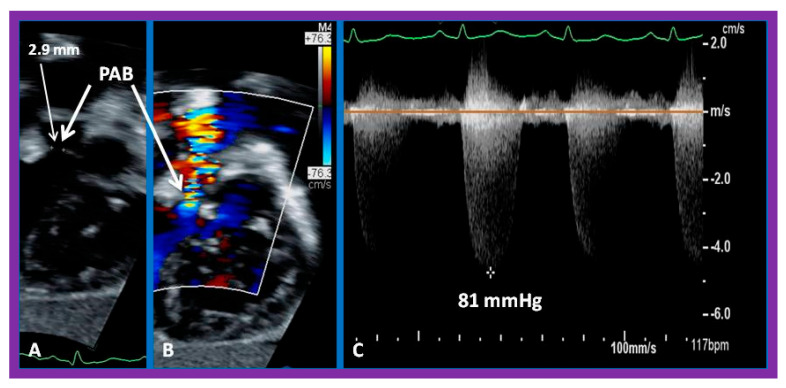
Echo–Doppler recordings demonstrating pulmonary artery banding (PAB) with a narrow two-dimensional width of 2.9 mm (**A**) and by color Doppler imaging (**B**). A significant pressure gradient of 81 mm Hg was recorded by CW Doppler (**C**). Reproduced from [21].

**Figure 5 children-10-00739-f005:**
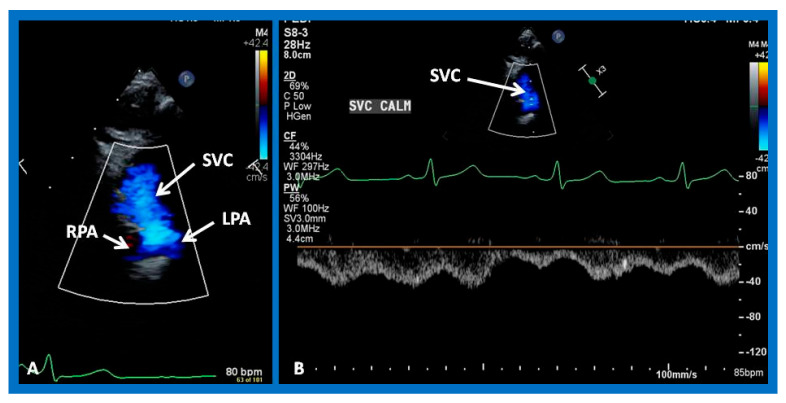
Selected video images from the suprasternal notch view demonstrating bidirectional Glenn anastomosis. The superior vena cava (SVC) emptying into the left (LPA) and right (RPA) pulmonary arteries by color flow imaging (**A**). Normal Doppler flow velocities across the Glenn shunt (**B**) suggests that the anastomotic site is not occluded. Reproduced from [21].

**Figure 6 children-10-00739-f006:**
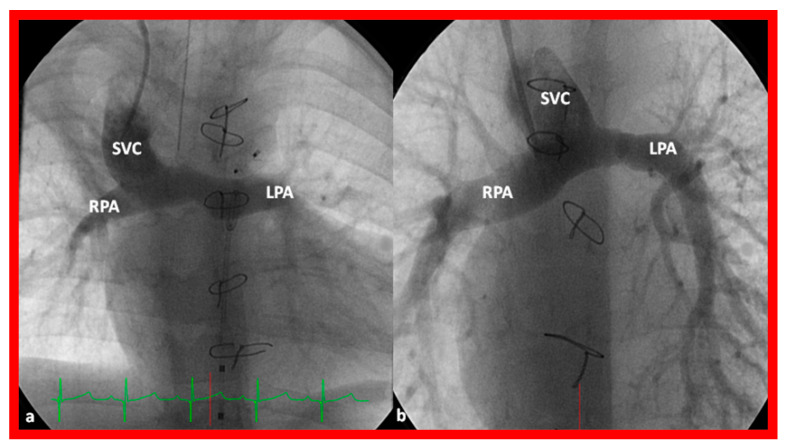
Selected cine frames demonstrating a bidirectional Glenn anastomosis in two different patients, (**a**,**b**). Unrestricted blood flow from the superior vena cava (SVC) to the left (LPA) and right (RPA) pulmonary arteries is seen. Sternal wires from a previous operation are visualized (not marked). Reproduced from [63].

**Figure 7 children-10-00739-f007:**
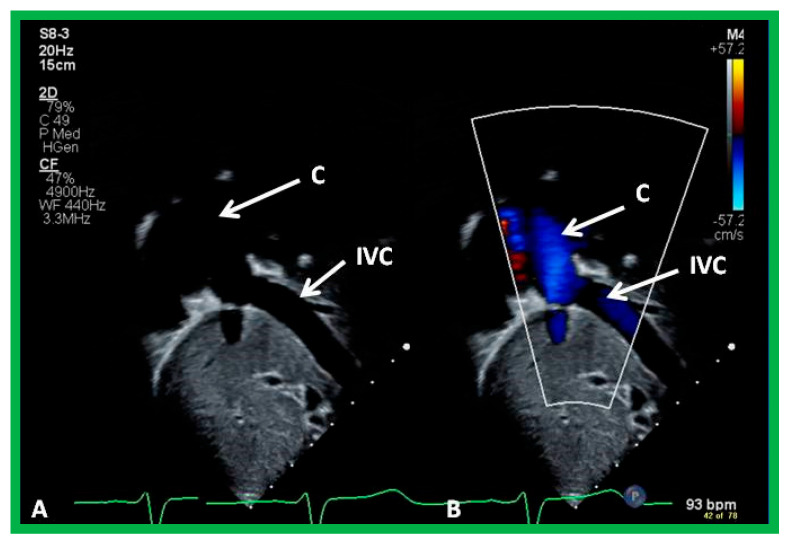
Selected video images by two-dimensional (**A**) and color echo–Doppler (**B**) illustrating anastomosis of the inferior vena cava (IVC) with a non-valved conduit (C). Note that the inferior vena cava to conduit connection is wide open. Reproduced from [21].

**Figure 8 children-10-00739-f008:**
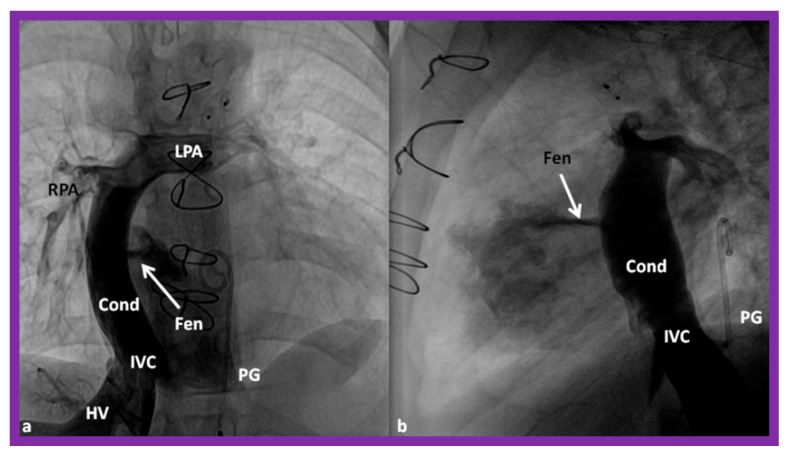
Selected cine frames in the posteroanterior (**a**) and lateral (**b**) projections, demonstrating Stage IIIA of the Fontan procedure redirecting the inferior vena caval (IVC) blood flow into the left (LPA) and right (RPA) pulmonary arteries via a non-valved conduit (Cond). The flow of the angiographic material across the fenestration (Fen) is pointed out with arrows in both (**a**,**b**). The hepatic veins (HV) and pigtail catheter (PG) in the descending aorta are labeled. Modified from [63].

**Figure 9 children-10-00739-f009:**
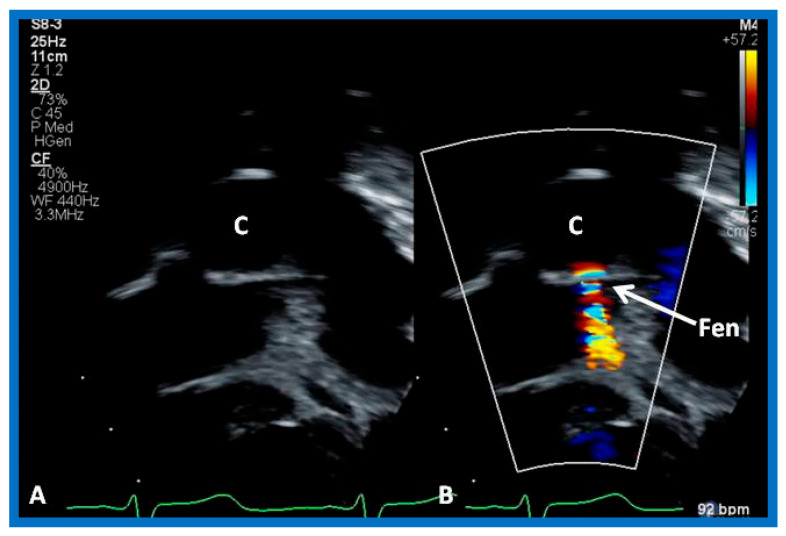
Echo–Doppler recordings in an apical four-chamber view by two-dimensional (**A**) and color Doppler (**B**) illustrating a cross-sectional view of the conduit (C) as shown in (**A**) and the fenestration (Fen) as demonstrated in (**B**). Turbulent flow is seen through the fenestration. Reproduced from [21].

**Figure 10 children-10-00739-f010:**
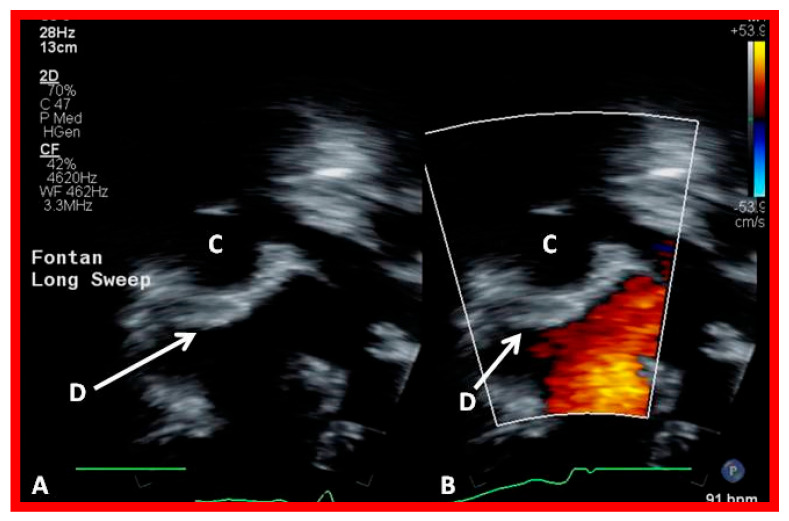
Echo–Doppler recordings in an apical four-chamber view illustrating the location of the Amplatzer Occluder device (D) (arrows in (**A**,**B**)). No residual shunt is demonstrated (**B**). Conduit (C) is labeled. Reproduced from [21].

**Figure 11 children-10-00739-f011:**
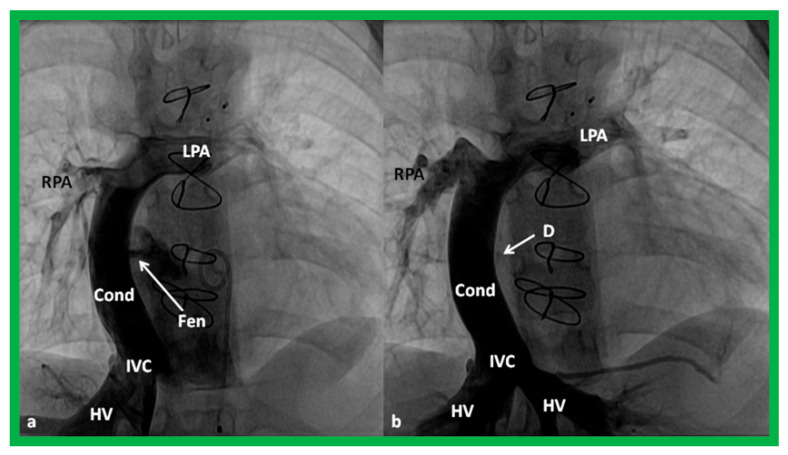
Cineangiographic images in the posteroanterior projection illustrating Stage IIIA of the Fontan operation transferring the blood from the inferior vena cava (IVC) into the pulmonary arteries via a non-valved conduit (Cond). The fenestration (Fen) is pointed out with an arrow in (**a**). The fenestration is occluded with an Amplatzer device (D), again, pointed out with an arrow in (**b**) (Stage IIIB). The hepatic vein (HV), left pulmonary artery (LPA), and right pulmonary artery (RPA) are marked. Reproduced from [63].

**Table 1 children-10-00739-t001:** Therapy of cardiac malposition.

Clinical Scenario	Treatment Options
**Neonates, infants, and children**	
Pulmonary oligemia	Prostaglandin E1 infusion, modified Blalock–Taussig anastomosis, stenting of the ductus arteriosus, and balloon pulmonary valvuloplasty, as deemed appropriate
Pulmonary plethora	Decongestive treatment, pulmonary artery banding
Total anomalous pulmonary venous return	Surgical connection of the common pulmonary vein and the left atrium
Pulmonary valvar stenosis	Balloon pulmonary valvuloplasty
Atrial septal defect	Device occlusion or surgical closure, as appropriate
Ventricular septal defect	Surgical repair
Patent ductus arteriosus	Device occlusion or surgical closure, as appropriate
Lesions with adequate-sized ventricles	Biventricular repair
Lesions with hypoplasia of the right or left ventricle	Staged total cavopulmonary connection (Fontan)
Patients with decreased splenic function (asplenia and polysplenia)	Antibiotic prophylaxis and Ladd’s procedure (if appropriate)
Interstage issues and postintervention complications	As summarized in Section 6 and Section 7
**Adults**	
Coronary artery disease	Percutaneous coronary intervention or surgical coronary artery bypass, as deemed appropriate
Supraventricular arrhythmias	Antiarrhythmic medications, catheter ablation (cryoablation or radiofrequency (RF) ablation), Maze procedure, as deemed appropriate
Complete atrioventricular block or sick sinus syndrome	Pacemaker therapy (dual-chamber pacemaker or leadless pacemaker implantation)
Prevention of sudden death	Implantation of a cardioverter–defibrillator (ICD) or subcutaneous ICDs, as appropriate
Prevention of embolic episodes	Initial anticoagulation and subsequent occlusion of the left atrial appendage
Aortic valve disease	Balloon aortic valvuloplasty or repair by surgery in younger patients. Transcatheter or surgical replacement of the aortic valve in the elderly
Mitral valve disease	Surgical repair or replacement of the mitral valve, as appropriate
Hypertrophic cardiomyopathy	Betablocker therapy, surgical myectomy, or alcohol septal ablation, as appropriate

## Data Availability

Not applicable.

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
