# Peer review of "Therapy of Patients with Cardiac Malposition"

_children, 2023, doi:10.3390/children10040739_

Round 1
Reviewer 1 Report
This manuscript introduced to us the current surgical treatment of cardiac malformations, systematiclly and completely, is of some value. The main problem is that most of the references are old and need to be updated. Other specific questions are as follows:
1. Second paragraph of 4.1(management of cardiac at initial presentation). What are the long-term consequences of decreased pulmonary blood flow? Is there a different consequence between early and late correction for decreased pulmonary blood flow?? What are the precautions, and what are the latest research developments regarding decreased pulmonary blood flow?
2. Second paragraph of 4.1(management of cardiac at initial presentation). What are the long-term consequences of increased pulmonary blood flow? Is there a different consequence between early and late correction for decreased pulmonary blood flow?? What are the precautions, and what are the latest research developments regarding decreased pulmonary blood flow?
3. What kinds of cardiac malposition would cause increased or decreased pulmonary blood flow?
4. “7. Post-intervention Issues and 8. Management of Cardiac Malposition Patients During Adulthood”. That how to improve the long-term life quality of children with CHD is of great concern to pediatric cardiologists or surgeons, please update the recent progresses concerning this issue beyond surgery methods, especially patients who received Fontan operation.
5. “9. Summary and Conclusion”. The last sentence “It is concluded that most of the problems encountered in patients with dextroposition and dextrocardia can effectively be treated with the currently available therapeutic modalities” seems inappropriate here. Because most part of the article is about other cardiac abnormalities and little is about dextrocardia.
Author Response
The reviewer states: “This manuscript introduced to us the current surgical treatment of cardiac malformations, systematically and completely, is of some value” - Thank you.
The reviewer also comments: “The main problem is that most of the references are old and need to be updated.” The reason for the old references is the author’s intent to acknowledge the original contributions of the cardiologists/surgeons who have created this knowledge and invented these procedures, for example Blalock and Taussig (1945), Muller and Danimann (1952), Fontan and Baudet (1972), de Level and his colleagues (1981) and others. Where applicable, new references are added, as deemed appropriate.
The reviewer also comments on other specific questions:
- Second paragraph of 4.1 (management of cardiac at initial presentation). What are the long-term consequences of decreased pulmonary blood flow? Is there a different consequence between early and late correction for decreased pulmonary blood flow?? What are the precautions, and what are the latest research developments regarding decreased pulmonary blood flow? – The long-term consequences of decreased pulmonary blood flow are development of clubbing of fingers and toes, cerebrovascular accidents, relative anemia, severe polycythemia, paradoxical embolism, brain abscess, coagulation abnormalities, hyper-uricemia, gout, and uric acid nephropathy. The short-term consequences of decreased pulmonary blood flow are hypoxemia, progressive acidosis, and death. These are now incorporated into the revised manuscript.
- Second paragraph of 4.1(management of cardiac at initial presentation). What are the long-term consequences of increased pulmonary blood flow? Is there a different consequence between early and late correction for decreased pulmonary blood flow?? What are the precautions, and what are the latest research developments regarding decreased pulmonary blood flow? – The long-term consequences of increased pulmonary blood flow are development of pulmonary hypertension and pulmonary vascular obstructive disease while the short-term consequences are congestive heart failure and death. These are now incorporated into the revised manuscript.
- What kinds of cardiac malposition would cause increased or decreased pulmonary blood flow? This was addressed in the first paper “Rao PS, Rao NS. Diagnosis of dextrocardia with a pictorial rendition of terminology and diagnosis. Children 2022; 9(12): 1977. https://doi.org/10.3390/ children9121977”. This paper is a companion paper to that paper.
- “7. Post-intervention Issues and 8. Management of Cardiac Malposition Patients During Adulthood”. That how to improve the long-term life quality of children with CHD is of great concern to pediatric cardiologists or surgeons, please update the recent progresses concerning this issue beyond surgery methods, especially patients who received Fontan operation. – Regarding post-intervention Issues, these were summarized, and appropriate references were provided. Regarding Management of Cardiac Malposition Patients During Adulthood, a detailed discussion of each item was presented.
- “9. Summary and Conclusion”. The last sentence “It is concluded that most of the problems encountered in patients with dextroposition and dextrocardia can effectively be treated with the currently available therapeutic modalities” seems inappropriate here. Because most part of the article is about other cardiac abnormalities and little is about dextrocardia – I beg to disagree; the discussion is about the treatment of dextroposition and dextrocardia all along.
The author wishes to convey his thanks to the reviewer for diligent review and constructive criticism.
Reviewer 2 Report
The article discusses the therapy of patients with cardiac malposition, as well as the management of associated congenital heart disease. To improve the organization of the content, it may be helpful to create a table summarizing all pertinent topics (to visualize all relevant topics at a glance) and to discriminate the topics specifically directed to the issues/management regarding cardiac malposition from those associated with congenital heart disease in general.
While the information provided covers the field of cardiac malposition, the depth of the discussion has been somewhat dismissed in favor of broad coverage. By creating a table and discriminating between topics, the article can be restructured to improve clarity and focus on the specific issues and management strategies relevant to patients with cardiac malposition.
Author Response
- The reviewer states “The article discusses the therapy of patients with cardiac malposition, as well as the management of associated congenital heart disease. To improve the organization of the content, it may be helpful to create a table summarizing all pertinent topics (to visualize all relevant topics at a glance) and to discriminate the topics specifically directed to the issues/management regarding cardiac malposition from those associated with congenital heart disease in general.” A table is prepared and included in the revised manuscript, as recommended by the reviewer.
- The reviewer states “While the information provided covers the field of cardiac malposition, the depth of the discussion has been somewhat dismissed in favor of broad coverage. By creating a table and discriminating between topics, the article can be restructured to improve clarity and focus on the specific issues and management strategies relevant to patients with cardiac malposition.” Again, a table is prepared and included in the revised manuscript and attempt was made to improve clarity, as suggested by the reviewer.
Round 2
Reviewer 1 Report
Thank you for the authors’ responses.
In response to Q1/2
I have one more question: Does delayed or early treatment of decreased or increased pulmonary blood flow affect alveolar development or long-term quality life of children? Because more than 90% of the alveoli are formed between 0 and 7 years of age (Nat Rev Dis Primers 5: 78,2019. doi:10.1038/s41572-019-0127-7.), and insufficient pulmonary blood perfusion can cause alveolar dysplasia and pulmonary dysfunction in adult (Circulation. 1977;56:647-651)?
Author Response
The reviewer asks “Does delayed or early treatment of decreased or increased pulmonary blood flow affect alveolar development or long-term quality life of children? Because more than 90% of the alveoli are formed between 0 and 7 years of age (Nat Rev Dis Primers 5: 78,2019. doi:10.1038/s41572-019-0127-7.), and insufficient pulmonary blood perfusion can cause alveolar dysplasia and pulmonary dysfunction in adult (Circulation. 1977;56:647-651)?”; yes, the answer is “yes”. I have added a sentence in the revised script to address the reviewer’s comments.
Reviewer 2 Report
The revised version of the paper has become easier to understand and better structured. The authors have adequately addressed my comments, with the exception of one point: I was unable to find any tables that provide a comprehensive summary of the points described in the paper.
Author Response
The reviewer states “The revised version of the paper has become easier to understand and better structured. The authors have adequately addressed my comments, with the exception of one point: I was unable to find any tables that provide a comprehensive summary of the points described in the paper.” – Thanks; the table was inadvertently forgotten to be included in the manuscript. It is now included in the revised script.